# Co-Inhibition of the DNA Damage Response and CHK1 Enhances Apoptosis of Neuroblastoma Cells

**DOI:** 10.3390/ijms20153700

**Published:** 2019-07-29

**Authors:** Kiyohiro Ando, Yohko Nakamura, Hiroki Nagase, Akira Nakagawara, Tsugumichi Koshinaga, Satoshi Wada, Makoto Makishima

**Affiliations:** 1Department of Clinical Diagnostic Oncology, Showa University Clinical Research Institute for Clinical Pharmacology and Therapeutics, 6-11-11 kita-karasuyama, setagaya-ku, Tokyo 157-8577, Japan; 2Department of Biomedical Sciences, Division of Biochemistry, Nihon University School of Medicine, 30-1 Oyaguchi-kamicho, Itabashi-ku, Tokyo 173-8610, Japan; 3Chiba Cancer Center Research Institute, Chiba 260-8717, Japan; 4Division of Molecular Medicine, Life Science Research Institute, Saga Medical Center Koseikan, Saga 840-8571, Japan; 5SAGA HIMAT Foundation, Saga 841-0071, Japan; 6Department of Pediatric Surgery, Nihon University School of Medicine, Tokyo 173-8610, Japan

**Keywords:** CHK1, ATM, DNA-PK, neuroblastoma, checkpoint abrogation

## Abstract

Checkpoint kinase 1 (CHK1) is a central mediator of the DNA damage response (DDR) at the S and G2/M cell cycle checkpoints, and plays a crucial role in preserving genomic integrity. CHK1 overexpression is thought to contribute to cancer aggressiveness, and several selective inhibitors of this kinase are in clinical development for various cancers, including neuroblastoma (NB). Here, we examined the sensitivity of MYCN-amplified NB cell lines to the CHK1 inhibitor PF-477736 and explored mechanisms to increase its efficacy. PF-477736 treatment of two sensitive NB cell lines, SMS-SAN and CHP134, increased the expression of two pro-apoptotic proteins, BAX and PUMA, providing a mechanism for the effect of the CHK1 inhibitor. In contrast, in NB-39-nu and SK-N-BE cell lines, PF-477736 induced DNA double-strand breaks and activated the ataxia telangiectasia mutated serine/threonine kinase (ATM)-p53-p21 axis of the DDR pathway, which rendered the cells relatively insensitive to the antiproliferative effects of the CHK1 inhibitor. Interestingly, combined treatment with PF-477736 and the ATM inhibitor Ku55933 overcame the insensitivity of NB-39-nu and SK-N-BE cells to CHK1 inhibition and induced mitotic cell death. Similarly, co-treatment with PF-477736 and NU7441, a pharmacological inhibitor of DNA-PK, which is also essential for the DDR pathway, rendered the cells sensitive to CHK1 inhibition. Taken together, our results suggest that synthetic lethality between inhibitors of CHK1 and the DDR drives G2/M checkpoint abrogation and could be a novel potential therapeutic strategy for NB.

## 1. Introduction

Transformed and untransformed cells respond to threats to genomic integrity, such as double-strand breaks (DSBs), by activating the DNA damage response (DDR), which enables DNA repair at the G1/S, S, and/or G2/M cell cycle checkpoints, thereby allowing progression through the cell cycle. The DDR is composed of two major signaling cascades, namely, the ataxia telangiectasia mutated serine/threonine kinase (ATM)-checkpoint kinase 2 (CHK2) cascade and the ataxia telangiectasia mutated and Rad3-related serine/threonine kinase (ATR)-checkpoint kinase 1 (CHK1) cascade. In general, DSBs activate the ATM-CHK2 cascade, whereas single-strand breaks are recognized and repaired by the ATR-CHK1 cascade [1]. Crosstalk between the cascades occurs at the levels of the downstream effectors of CHK1 and CHK2 [2], and the pathways ultimately converge to activate the well-established DNA repair mechanisms, homologous recombination and non-homologous end joining (NHEJ) [3].

CHK1 and CHK2 are serine/threonine kinases that regulate the activity of their target proteins, including p53 and CDC25A–C phosphatases, via phosphorylation. In turn, the CDC25 proteins regulate the activity of cyclins and the cyclin-dependent kinases (CDK). CHK1 is principally responsible for initiating repair during S-phase replication (the replication checkpoint) and G2/M (the mitotic entry checkpoint) via phosphorylation of the CDC25 family members and the serine/threonine kinase Wee1, whereas CHK2 coordinates repair for S-phase entry (G1 checkpoint) via phosphorylation of p53, which transactivates the CDK inhibitor CDKN1A (also known as p21) [4].

A large body of evidence has shown that impairment of the DDR plays a pivotal role in tumorigenesis, tumor progression, and therapy resistance resulting from increased genomic instability [5,6]. Aberrant CHK1 expression has been frequently observed in various cancers and is thought to contribute to malignant phenotypes, making it an attractive therapeutic target [7]. Indeed, more than a decade of research has resulted in the development of many small-molecule inhibitors of CHK1 (hereafter referred to as CHK1is) and of other DDR-related kinases [8,9]. The rationale for targeting CHK1 and other checkpoint kinases comes from the concept of cancer cell vulnerability to “checkpoint abrogation.” The genomic instability of cancer cells is mostly attributable to defects in the G1 checkpoint resulting from functional deficiency of the tumor suppressors p53 or Rb. Thus, blockade of the G2/M checkpoint by CHK1is (i.e., checkpoint abrogation) would prevent cancer cells from repairing critical DNA insults, thereby compromising their survival and/or sensitizing them to other chemotherapeutic agents while leaving normal cells unaffected [10,11,12]. The sensitivity of cancer cells to CHK1is has been correlated with endogenous replication stress (RS) resulting from oncogene-induced hyperproliferation [13]. Consistent with this, previous studies have reported that neuroblastomas (NBs) with MYCN amplification, the most common diagnostic marker of NB with unfavorable histology, are sensitive to CHK1is [14,15]. One potential explanation for this sensitivity is that MYCN-mediated RS might upregulate CHK1 expression and activation, rendering the cells addicted to CHK1-mediated DDR for their survival and, hence, vulnerable to CHK1is. Preclinical evaluation of multiple types of CHK1i have confirmed their antiproliferative effects in various cancers, although only a few CHK1is have progressed to early phase clinical trials [16,17,18].

In this study, we investigate the potential utility of a CHK1i for the therapy of NB. We found that MYCN-amplified human NB cell lines classified as having low sensitivity to CHK1is activated the ATM-p53-p21 pathway of the DDR upon CHK1i treatment, thus enabling their survival. However, combination treatment with the CHK1i and an inhibitor of either ATM or DNA-PK, a well-known sensor for the DDR, efficiently inhibited the growth of CHK1i-insensitive NB cells. These results indicated that co-inhibition of CHK1 and the DDR is a potential novel therapeutic option for NB.

## 2. Materials and Methods

### 2.1. Cell Culture and Treatments

The human NB cell lines CHP134, NB-39-nu, SKN-BE, and SMS-SAN were obtained from the American Type Culture Collection (Manassas, VA, USA) and the RIKEN Bioresource Cell Bank, Tohoku University Cell Resource Center (Miyagi, Japan) and grown in RPMI1640 medium supplemented with 10% heat-inactivated fetal bovine serum, 50 µg/mL penicillin, and 50 µg/mL streptomycin (Thermo Fisher Scientific, Waltham, MA, USA). CHK1i (PF-477736), ATMi (Ku55933), and DNA-PKi (NU7441) were obtained from Sigma-Aldrich St. Louis, MO, USA. Stock solutions were made in dimethyl sulfoxide (DMSO).

### 2.2. Cell Viability Assays and Morphological Analysis

Two assays were performed. For the Trypan blue exclusion test of cell viability, cells (5 × 10^5^ cells/well) were seeded in 24-well plates. After incubation for the indicated time periods, cells were trypsinized. The suspended cells were mixed with Trypan blue (Sigma) and the number of cells with a clear cytoplasm was counted. For the alamarBlue assay, cells (2500 cells/well) were seeded in 96-well plates and incubated as described for the indicated times. The alamarBlue Cell Viability Regent (Invitrogen, Carlsbad, CA, USA) was added to the cells for 1 h, and immunofluorescence was measured using a SpectraMax M5e (Molecular Devices). Alternatively, cells (2.5 × 10^4^ cells/well) were incubated in 6-well plates for 6 days, and then stained with May-Grunwald Giemsa stain (Muto Pure Chemicals) and analyzed by microscopy. The medium with or without CHK1i (PF-477736) was changed by every 3 days.

### 2.3. Microarray Analysis of Gene Expression

Cells were treated as described and harvested. Total RNA was prepared using an RNeasy mini kit (Qiagen, Hilden, Germany) according to the manufacturer’s procedure. Total RNA was supplied to a contract research organization (Takara, Kusatsu, Japan) for analysis using an Agilent SurePrint G3 Human GE 8 × 60 K Microarray. Gene expression data were analyzed with paired *t*-tests and the fold change in each transcript was calculated using GeneSpring GX12 software (Agilent Technologies, Santa Clara, CA, USA).

### 2.4. Quantitative Reverse-Transcription PCR

Total RNA was reverse transcribed using random primers and Superscript II (Invitrogen, Carlsbad, CA, USA) according to the manufacturer’s instructions. cDNA was amplified by real-time PCR to quantify expression of p53, p21, MDM2, and ACTB mRNA. MDM2 and ACTB TaqMan probes (Hs01066930_m1 and Hs01060665_g1, respectively) were purchased from Applied Biosystems. p53 and ACTB mRNA expression was measured using SYBR Green reagent-based real-time PCR. qPCR was performed using a StepOnePlus Real-Time PCR System with TaqMan Fast Universal PCR Master Mix or Power SYBR Green PCR Master Mix (Applied Biosystems) according to the manufacturer’s instructions. The specific primers were: p53, 5′-GGTTTCCGTCTGGGCTTCTT-3′ and 5′-CCTCCGTCATGTGCTGTGAC-3′; ACTB, 5′-GACAGGATGCAGAAGGAGAT-3′ and 5′-GAAGCATTTGCGGTGGACGAT-3′; p21, 5′-AGACTCTCAGGGTCGAAAAC-3′ and 5′-TAGGGCTTCCTCTTGGAGAA-3′.

### 2.5. Immunoblotting

Whole cell lysates were prepared in 1% NP-40 cell lysis buffer (Boston BioProducts, Ashland, MA, USA) with Complete Mini Protease Inhibitor Cocktail (Roche, Basel, Switzeland) and phosphatase inhibitor cocktail (PhosSTOP, Sigma Aldrich). Whole cell extracts (25–100 μg protein) were separated by SDS-PAGE using an XCell SureLock Mini-Cell with Bis-Tris gels or Tris-acetate gels in MOPS, MES, or Tris-acetate SDS running buffer (Invitrogen), according to the manufacturer’s instructions. Samples were transferred to nitrocellulose membranes (Bio-Rad, Hercules, CA, USA) and blocked with 5% bovine serum albumin (BSA) in Tris-buffered saline with 0.1% Tween (TBST). Membranes were then probed overnight at 4 °C with the following primary antibodies diluted in TBST/5% BSA: monoclonal anti-p53 (DO-1, Santa Cruz Biotechnology; 1:500, Dallas, TX, USA), monoclonal anti-CHK1 (G-4, Santa Cruz Biotechnology; 1:500), monoclonal anti-β-actin (AC-74, Sigma; 1:4000), monoclonal anti-MDM2 (SMP14, Santa Cruz Biotechnology 1:500), monoclonal anti-p21 Waf1/Cip1 (12D1), monoclonal anti-ATM (D2M2), monoclonal anti-phospho (p)-ATM-Ser1981 (D25E5), monoclonal anti-p-CHK1-Ser345 (133D3), polyclonal anti-p-p53-Ser15, polyclonal anti-p-Histone H2A.X-Ser139, polyclonal anti-p-Histone H3-Ser10, polyclonal anti-caspase-3, and polyclonal anti-poly (ADP-ribose) polymerase (Cell Signaling Technology; 1:1000, Danvers, MA, USA). Membranes were washed in TBST and probed with horseradish peroxidase-coupled anti-rabbit or anti-mouse secondary antibodies (Cell Signaling Technology) diluted 1:2500 in TBST/5% ECL Blocking Agent (GE Healthcare, Chicago, IL, USA) for 1 h. Finally, the membranes were washed and developed with ECL Western Blotting Detection Reagents (GE Healthcare). Bands of interest were detected with an ImageQuant LAS 500 (GE Healthcare). The band intensity was normalized to β-actin intensity and measured using Quantity one software (Bio Rad).

### 2.6. Immunofluorescence Microscopy

Cells were grown on coverslips, fixed with 3.7% formaldehyde in PBS for 15 min, washed twice with PBS for 5 min, and then permeabilized with PBS/0.2% Triton X-100 for 10 min. Cells were then incubated twice with PBS/0.01% Triton X-100 for 5 min, blocked with PBS/3% BSA/0.01% Triton X-100 for 1 h, and incubated with polyclonal antibodies against p-p53-Ser15 or p-Histone H2A.X-Ser139 antibodies (Cell Signaling Technology) at 1:1000 in PBS/1% BSA/0.01% Triton X-100. Cells were then washed twice with PBS/0.01% Triton X-100 for 5 min and incubated with Alexa Fluor 568-conjugated goat anti-mouse IgG (Molecular Probes, Eugene, OR, USA) at 1:500 in PBS/1% BSA/0.01% Triton X-100. Cells were mounted with Vectashield Mounting Medium containing 4′,6-diamidino-2-phenylindole (Vector Laboratories) and visualized and photographed using a FLUOVIEW FV10i confocal fluorescence microscope (Olympus, Shinjuku, Japan).

### 2.7. Flow Cytometry

Floating and attached cells were collected, treated with 500 µg/mL RNase A (Sigma), and then incubated with 50 µg/mL of propidium iodide (Sigma) in 0.2% Triton X-100 for 30 min at room temperature. The DNA content of 10,000 cells per sample was analyzed using a FACSCalibur flow cytometer (Becton Dickinson, Franklin Lakes, NJ, USA).

### 2.8. RNA Interference

Stealth RNAi siRNA specific for p53 (TP53HSS110905, 186390 and 186391) and a negative control sequence (medium GC-content) were obtained from Thermo Fisher Scientific. Cells were transiently transfected with 10 nM of siRNA using Lipofectamine 3000 (Invitrogen), and whole cell lysates were prepared 48 h after transfection.

### 2.9. Statistical Analysis

Statistical analysis was performed using Microsoft Excel. Data were analyzed using unpaired Student’s *t*-test. The results are presented as the mean ± standard deviation (SD).

## 3. Results

### 3.1. CHK1 Inhibition Activates Downstream Targets of p53 in NB Cells

MYCN-amplified NB tumor specimens express higher levels of CHK1 mRNA than non-MYCN-amplified NB cells [14], suggesting that the sensitivity of such NB cells to CHK1 inhibition may be related to high levels of MYCN-induced RS. We analyzed CHK1 mRNA levels in public datasets of NB patient samples using R2, the Genomics Analysis and Visualization Platform (https://hgserver1.amc.nl/cgi-bin/r2/main.cgi) and confirmed that high CHK1 expression significantly correlated with an unfavorable prognosis in NB patients (*n* = 88, *p* < 0.01). CHK1 and MYCN expression were also significantly correlated in these samples (*R^2^* = 0.57, *p* < 0.01; Appendix A).

To investigate the sensitivity of human NB cell lines to CHK1 inhibition, we examined the effects of the CHK1i PF-00477736 on the proliferation of four MYCN-amplified NB cell lines: NB-39-nu, SMS-SAN, CHP134, and SK-N-BE [19,20,21,22]. PF-00477736 was originally identified as a potent, selective ATP-competitive small-molecule inhibitor of CHK1 (*Ki* = 0.49 nM) that potentiates the cytotoxic effect of conventional chemotherapeutic agents in vitro and in vivo [23,24]. We found that CHP134 and SMS-SAN cells were much more sensitive to 1 μM PF-477736 compared with SK-N-BE and NB-39-nu cells, as demonstrated by assessment of the proliferation assay for 3 days (Figure 1A). Further, IC_50_ analysis was performed on these cell lines to confirm their sensitivity to PF-477736 (Appendix A). To examine the potential molecular changes underlying CHK1i sensitivity, we performed a microarray analysis to identify genes differentially expressed in SMS-SAN and NB-39-nu cells, which showed high and low sensitivity to PF-477736, respectively, after treatment with or without 1 μM PF-477736. Among the genes most differentially expressed in the two cell types were two pairs of p53 target genes. After incubation with PF-477736, SMS-SAN cells showed upregulated expression of BAX and PUMA, both of which are pro-apoptotic proteins, whereas NB-39-nu cells showed upregulation of p21, a CDK inhibitor, and MDM2, a negative regulator of p53 (Figure 1B). Because MYCN has been suggested to transcriptionally upregulate p53 in NB [25], we assessed the expression of MYCN, p53, and CHK1 in these cell lines by immunoblotting. Consistent with their relative sensitivity to CHK1i, CHP134 and SMS-SAN cells expressed higher MYCN levels than did either of the more insensitive cell lines, SK-N-BE and NB-39-nu, whereas CHK1 expression was relatively lower in NB-39-nu cells among the four lines (Figure 1C). Interestingly, p53 expression tended to correlate inversely with that of MYCN, with the cells exhibiting lower sensitivity to CHK1is expressing higher p53 levels (Figure 1C). These results suggest that increased p53 protein levels may be associated with the reduced sensitivity to CHK1is of MYCN-amplified NBs.

### 3.2. CHK1 Inhibition Upregulates the ATM-p53 Axis in NB Cells

To determine whether the upregulation of p21 and MDM2 in CHK1i-treated NB-39-nu cells was p53 dependent, we performed siRNA-mediated knockdown (KD) of p53 and examined p21 and MDM2 expression by RT-qPCR. CHK1i (1 μM) treatment increased p21 and MDM2 mRNA levels, as expected, but the upregulation was significantly blunted by p53 KD (Figure 2A). Moreover, immunoblotting (Figure 2B) and immunofluorescence staining (Figure 2C) showed that levels of active p53, phosphorylated on Ser15, were dramatically elevated in CHK1i-treated NB-39-nu cells compared with control cells (Figure 2C), although p53 transcripts was significantly downregulated by the CHK1i treatment (*p* < 0.05). p53 phosphorylation on Ser15, which is mediated by ATM, increases its stability and transactivity in response to DNA damage, especially in the presence of DSBs [26]. In agreement with our microarray and RT-qPCR analyses, p21 and MDM2 protein levels were also markedly upregulated by CHK1i treatment of NB-39-nu cells (Figure 2B). Notably, these events were accompanied by increasing levels of active p-ATM-Ser1981 (Figure 2B). In contrast, the CHK1i-sensitive SMS-SAN cell line showed a lesser extent of increasing levels of phosphorylated on Ser15 and p21 compared to NB-39-nu. Strikingly, decreased levels of active p-ATM-Ser1981 were observed in SMS-SAN cells (Appendix A). Taken together, these data suggested that CHK1i treatment in CHK1i-insenstive NB cells might stabilize the p53 protein through activation of the ATM-p53-p21 axis-mediated DDR.

### 3.3. CHK1 Inhibition Induces DSB-Stimulated DDR Signaling

To confirm that CHK1i activates the ATM-p53 signaling axis by inducing DSBs at the doses employed here, we examined expression of the histone variant γH2AX, which is a marker of DSBs. Indeed, γH2AX levels were markedly upregulated in NB-39-nu cells exposed to 1–10 μM CHK1i (PF-477736), and a similar pattern of dose dependence was observed for p-ATM-Ser1981, p-p53-Ser15, and p21 levels (Figure 3A). In keeping with the inverse dose dependence, flow cytometric analysis of the cell cycle distribution revealed that 5 and 10 μM of CHK1i (PF-477736) resulted in massive apoptotic cell death, as detected by the increasing size of the sub-G1 population (Figure 3B) [27]. Furthermore, cell viability analysis showed that the increasing size of the sub-G1 population is likely to be associated with growth inhibiting effect by CHK1i (PF-477736) (Appendix A). Consistent with previous studies, high levels of basal γH2AX were observed in the CHK1i-sensitive SMS-SAN cell line compared to the NB-39-nu cell line [28]. These results indicate that CHK1i induced DSBs might trigger the DNA repair pathway through ATM-p53-p21 for cell survival in the CHK1i-insensitive NB-39-nu cell line. However, exposure of NB-39-nu cells to 5 and 10 μM CHK1i (PF-477736) were excessively cytotoxic, and thus DNA repair machinery might be unable to prevent cell death.

### 3.4. Co-Targeting of ATM and DNA-PK Overcomes CHK1 Inhibitor Insensitivity and Induces Apoptosis in NB Cells

Next, we asked whether the insensitivity of NB-39-nu cells to CHK1i could be overcome by concomitant pharmacological inhibition of ATM-initiated DDR pathways. To this end, NB-39-nu cells were incubated with a specific ATMi, Ku55933, in the presence or absence of PF-477736. Indeed, whereas treatment with either the CHK1i at 0.175 μM (below its IC_50_) or 10 μM ATMi had modest effects on cell viability, the combination of both inhibitors significantly decreased cell viability compared with either single agent alone (Figure 4A), indicating that blockade of ATM increased the sensitivity of NB-39-nu cells to the CHK1i. Consistent with this finding, levels of the apoptosis markers cleaved caspase-3 and cleaved poly (ADP-ribose) polymerase were upregulated to a much greater extent in the presence of the CHK1i plus ATMi compared with either one alone (Figure 4B). Importantly, expression of the mitosis marker p-histone H3 was detected only in cells treated with both CHK1i and ATMi, suggesting that these inhibitors induced apoptosis in M phase of the cell cycle. Apparent cell shrinkage and disruption of cell–cell contact (Figure 4C), and an increase in the proportion of cells in sub-G1 and mitosis (Appendix A), which confirms that apoptosis is occurring in mitosis—a phenomenon known as mitotic catastrophe, was only observed under the condition of combination treatment. Moreover, while cell treatment with 0.175 μM CHK1i or the ATMi had minimal effects on DSBs, as detected by γH2AX protein expression, the same doses in combination caused a marked increase in DSBs (Figure 4B). Surprisingly, however, the p53-p21 axis appeared to be incompletely blocked, even in the presence of both inhibitors, suggesting the existence of an alternative, ATM-independent pathway between DSBs and p53. It is also possible that detection of p-ATM-Ser1981 might not faithfully reflect the activity of the ATM-initiated DSB repair pathway in this experimental context, because p-ATM-Ser1981 expression was still elevated in NB-39-nu cells treated with the combination of CHK1i plus ATMi (Figure 4B). This apparent discrepancy between Ser1981 phosphorylation and ATM kinase activity has been reported by others [29] and remains to be elucidated.

Because the ATMi enhanced CHK1i-induced DSBs and growth inhibition in NB-39-nu cells, we next asked whether the same effect could be obtained by inhibition of other DDR mechanisms. To this end, we treated NB-39-nu cells with CHK1i in combination with NU7441, a selective inhibitor of DNA-PK, which is responsible for NHEJ in DNA repair. Intriguingly, combined inhibition of both CHK1 and DNA-PK not only significantly decreased cell viability compared with either single agent alone, but also had a greater growth-inhibitory effect than the combination of CHK1i and ATMi (*p* < 0.05, Figure 5A). Finally, we confirmed that these effects were not restricted to the NB-39-nu cell line by assessing the effects of the inhibitors on the second CHK1i-insensitive NB cell line, SK-N-BE. Indeed, combination treatment with the CHK1i and ATMi or DNA-PKi significantly reduced the viability of SKN-BE cells to a greater extent than did the single agents alone (*p* < 0.05, Figure 5B). Taken together, the results presented here strongly suggest that inhibition of DDR-related proteins can overcome CHK1i insensitivity and inhibit the growth of MYCN-amplified NB cells.

## 4. Discussion

In the present study, we evaluated the growth-inhibitory effects of the CHK1i PF-477736, as well as the underlying molecular events, in MYCN-amplified NB cell lines with varying degrees of CHK1i sensitivity. We found that high CHK1i concentrations induced DSBs and DDR pathway activation, suggesting that sensitivity to CHK1i might depend not only on MYCN amplification but also on DDR features specific to each cell line. In accordance with this hypothesis, we found that pharmacological inhibition of ATM or DNA-PK overcame the CHK1i insensitivity of the NB cell lines.

Recent studies have shown that CHK1i alone can inhibit the proliferation of certain cancers, including MYCN-amplified NB [30,31,32,33,34]. Clinical studies of CHK1i monotherapy are still ongoing, but thus far, it has been investigated in advanced squamous cell carcinoma of the head and neck, lung cancer, and recurrent high-grade serous or endometrioid ovarian cancer, although only limited benefit was observed [16,17]. Therefore, we propose that further preclinical studies should focus on determining the features critical for tumor cell survival in the presence of CHK1i and on identifying the agents best able to complement the anti-tumor effects of these inhibitors.

Increasing evidence has indicated that synthetic lethality using ATR, CHK1, or Wee1 inhibitors can be used to exploit oncogene-induced RS in cancer [18,35,36,37]. During cancer development, a driver oncogene promotes RS, exemplified by aberrant replication origin licensing or firing compared with normal cells [38,39]. Inhibitors that induce checkpoint abrogation in the S phase of the cell cycle, such as CHK1i, could thus induce stalled replication forks and DNA breaks, leading to apoptosis. In fact, oncogene-induced RS has been suggested to be a hallmark of vulnerability to the checkpoint abrogation strategy of cancer therapy [40]. A very recent study reported that treatment with ATR plus Wee1 inhibitors led to tumor remission and inhibited metastasis with minimal side effects in an orthotopic breast cancer model [41]. Thus, the results of proof-of-concept clinical trials to determine the safety and initial efficacy profiles of the various combination therapies are eagerly awaited.

Several markers of endogenous RS have been proposed to predict cellular sensitivity to CHK1i, including mutations in *CDKN2A* or *RB1* genes, amplification of *CCNE1* and *MYC* family genes, increased basal p-CHK1-S296 levels, genomic deletions in CDKN2A/p16, and increased p-RPA2-Ser4/8 foci [18,30,33,42]. Further evaluation of these candidates will be necessary to define their utility as predictive markers for CHK1i sensitivity in clinical trials.

Consistent with previous reports, we observed an increased level of DSBs in NB cell lines treated with CHK1i [43,44]. If DSB formation is required for CHK1i-induced cell death, it raises the question: What are the potential advantages of CHK1i over traditional cytotoxic chemotherapies? CHK1is were proposed to be potential chemosensitizers, but early phase II studies were unsuccessful for various reasons, including unacceptable side effects [45,46,47]. We speculate that this may at least partly be due to enhanced cytotoxicity resulting from CHK1i-induced DSBs, in combination with a DNA-damaging chemotherapeutic agent. Several checkpoint kinase inhibitors, including second-generation CHK1is, are currently in phase II trials in combination with chemotherapy and radiotherapy. The results of these trials will be important in determining the future direction of the checkpoint abrogation strategy for cancer therapy.

While most studies of the checkpoint abrogation strategy have focused on oncogene-induced RS, Chen et al. performed a synthetic lethal screen and identified deficiency of the Fanconi anemia (FA) DNA repair pathway as critical for cell sensitivity to CHK1i [48], which suggested that DNA repair-deficient tumors might be particularly vulnerable to CHK1i therapy. The results of the present study are consistent with this. We showed that inhibitors of ATM or DNA-PK, both prominent DNA damage sensors for initiating the DDR, reduced the viability of NB cells, even in the presence of low CHK1i concentrations that themselves had little to no effect on DSB induction. These results are clinically relevant because they suggest that combination therapy may reduce the toxic side effects of CHK1i on normal cells.

The development in recent years of highly selective kinase inhibitors has dramatically improved the success of cancer therapy. However, most of the targeted driver oncogenes are receptor tyrosine kinases [49], whereas the checkpoint kinases phosphorylate serine and threonine. The combination of oncogene-induced RS and DDR deficiency resulting in vulnerability to checkpoint kinase inhibitors could be a common feature of cancer. If so, such inhibitors could potentially be used to treat a wide spectrum of cancers. Chemotherapy has two main goals: One is to cure the disease and the other is to delay disease progression. In line with this, we suggest that checkpoint kinase inhibitors might be important as third or later stage chemotherapies to improve the patient’s quality of life, particularly those with tumors resistant to targeted therapy, such as receptor tyrosine kinase inhibitors. With this in mind, we included low concentrations of PF-477736 (<IC_50_ for inhibition of proliferation) to avoid toxicity, and we examined the effects of relatively chronic treatment (6–7 days). In the future, we plan to extend these studies to clarify the efficacy and toxicity of combination and longer duration therapy in vivo.

## 5. Conclusions

We found that CHK1i treatment induced DSBs in NB cells, which activated the ATM-p53 DDR pathway. Accordingly, combination blockade of CHK1 together with inhibitors of ATM or DNA-PK to block homologous recombination or NHEJ DNA repair, respectively, decreased the viability of CHK1i-insensitive cells regardless of the effects on p53 downstream targets (Figure 6). Thus, our findings suggest that blockade of the DDR could increase the potency of CHK1i and thus enable the checkpoint abrogation strategy of cancer therapy to be applied to a wide spectrum of cancers.

## Figures and Tables

**Figure 1 ijms-20-03700-f001:**
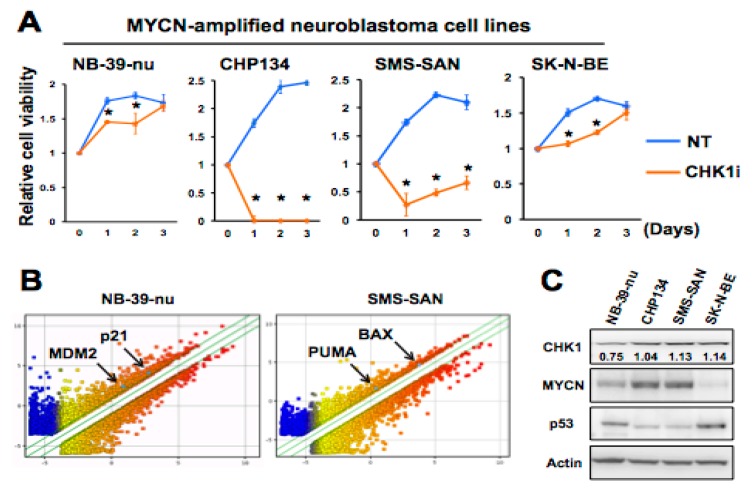
Checkpoint kinase 1 (CHK1) inhibition activates downstream targets of p53. (**A**) Cell viability assay of four MYCN-amplified neuroblastoma (NB) cell lines after exposure to dimethyl sulfoxide (DMSO) (NT) 1 μM CHK1 inhibitor (CHK1i) (PF-477736) for the indicated times. Data are presented as the mean ± SD of three independent experiments. * *p* < 0.05. (**B**) Microarray analysis of CHK1i-sensitive SMS-SAN cell line and the relatively insensitive NB-39-nu cell line at 36 h after treatment with 1 μM CHK1i or DMSO. (**C**) Immunoblot analysis of basal levels of CHK1, MYCN, and p53 in NB cells. β-actin was used as a loading control. Representative numbers were normalized to the intensity of the indicated bands.

**Figure 2 ijms-20-03700-f002:**
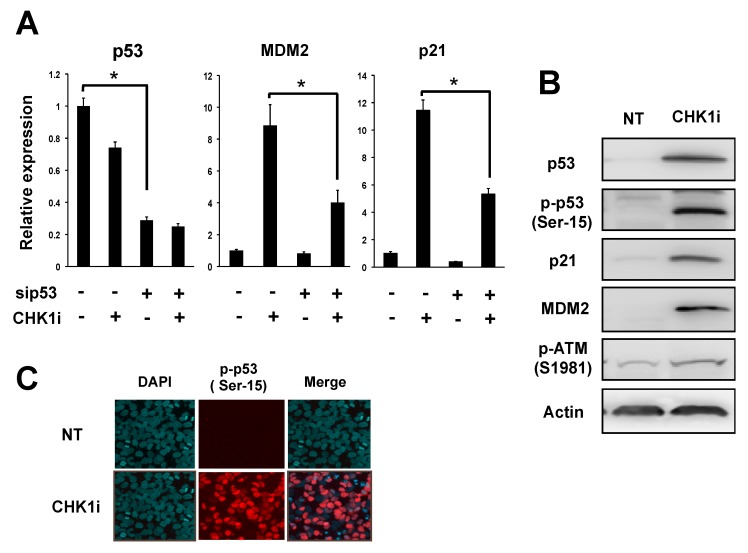
CHK1 inhibition activates the ataxia telangiectasia mutated serine/threonine kinase (ATM)-p53-p21 axis of the DNA damage response (DDR) pathway. (**A**) Quantitative RT-PCR analysis of p53, MDM2, and p21 mRNA levels in NB-39-nu cells transfected with 10 nM p53-specific or control siRNA for 24 h and then treated with 1 μM CHK1i (PF-47736) or DMSO for 24 h. Data are presented as the mean ± SD of triplicates and are normalized to *ACTB* levels. * *p* < 0.05. (**B**) Immunoblot analysis of the indicated proteins in NB-39-nu cells treated for 24 h with 1 μM CHK1i or DMSO (NT). β-actin was used as a loading control. (**C**) Indirect immunofluorescence of NB-39-nu cells treated for 24 h with 1 μM PF-477736 or DMSO. Cells were stained with polyclonal anti-p-p53-Ser15 (red) and 4′,6-diamidino-2-phenylindole (DAPI, nuclear DNA, blue).

**Figure 3 ijms-20-03700-f003:**
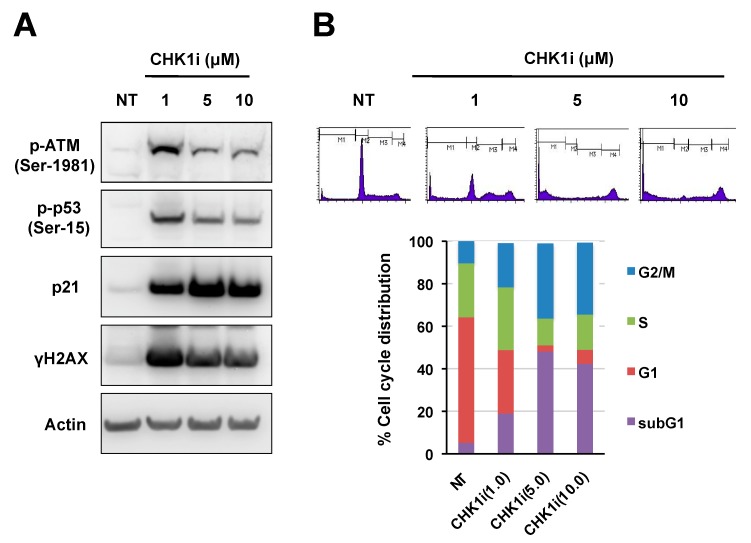
CHK1 inhibition induces double-strand break (DSB)-mediated DDR signaling. (**A**) Immunoblot analysis of the indicated proteins in NB-39-nu cells after treatment with DMSO (0, NT), 1, 5, or 10 μM CHK1i (PF-477736) for 24 h. (**B**) Flow cytometric cell cycle analysis of cells treated as for (**A**) and then stained with propidium iodide (PI) to assess the cell cycle distribution. Upper panels show the PI histograms; lower panels show the proportion of cells in each phase of the cell cycle after the indicated treatments.

**Figure 4 ijms-20-03700-f004:**
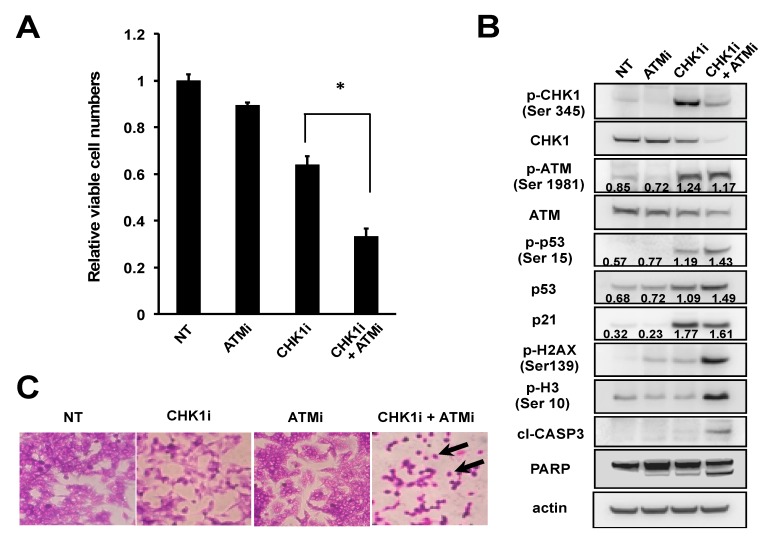
Combination treatment with CHK1 and ATM inhibitors induces NB-39-nu cell death. (A–C) NB-39-nu cells were treated with DMSO (NT), 0.175 μM PF-477736 (CHK1i), 10 μM Ku55933 (ATMi), or 0.175 μM PF-477736 plus 10 μM Ku55933 (CHK1i + ATMi). Cells were then harvested and analyzed for (**A**) alamarBlue cell viability assay, (**B**) protein expression by immunoblotting at 7 days after indicated treatments, or (**C**) Giemsa staining and microscopy at 6 days after indicated treatments. In (A), the graphs were plotted using the values from four independent experiments. Data are presented as the mean ± SD. * *p* < 0.05. cl-CASP3, cleaved caspase-3; PARP, poly (ADP-ribose) polymerase. In (B), Representative numbers were normalized to the intensity of the indicated bands. In C, black arrows indicate the examples of cell shrinkage with disruption of cell–cell contacts.

**Figure 5 ijms-20-03700-f005:**
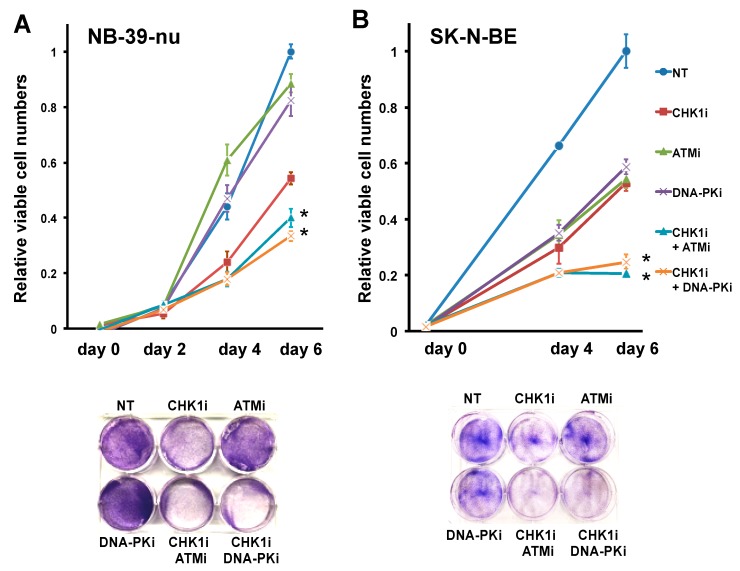
Combination treatment with either ATMi or DNA-PKi sensitizes NB cells to CHK1 inhibition. (**A**) CHK1i-insensitive NB-39-nu cells and (**B**) SK-N-BE cells were treated with DMSO (NT) or the indicated combinations of 0.175 μM CHK1i (PF-477736), 10 μM ATMi (Ku55933), and/or 1 μM DNA-PKi (NU7441) for 6 days. At the indicated times, cells were harvested and analyzed for cell viability (upper panels) or Giemsa staining (lower panel). Data are presented as the mean ± SD of four independent experiments. * *p* < 0.05.

**Figure 6 ijms-20-03700-f006:**
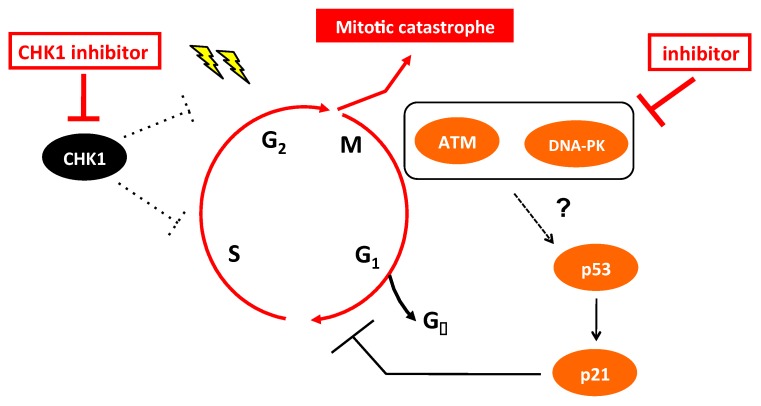
Blockade of the DDR enhances the therapeutic efficacy of CHK1i in MYCN-amplified NB cells.

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
