# Peer review of "Co-Inhibition of the DNA Damage Response and CHK1 Enhances Apoptosis of Neuroblastoma Cells"

_ijms, 2019, doi:10.3390/ijms20153700_

Round 1
Reviewer 1 Report
Ando et al. presented related experimental data to show co-inhibition of the DNA damage response and 2 CHK1 enhances apoptosis of neuroblastoma cells. In that respect, the authors suggested that combination treatment with the CHK1i and an inhibitor of either ATM or DNA-PK can inhibit the growth of CHK1i-insensitive NB cells.
Even though CHK1i is already used for early phase clinical trials and many other types of inhibitors were introduced in previous studies, in that respect of that the authors tried to check the effects of CHK1i in CHK1i-insensitive or -sensitive NB cell lines, this study is interesting. However, it looks that more detailed information is required in the text and additional experiments or analyses are required for supporting their conclusion.
Major comments;
1. There are many other CHK1 inhibitors including Ro-3306, AZD7762, Rabusertib, CHIR-124, MK-8776, GDC-0575 (ARRY-575, RG7741), CCT245737, Prexasertib HCl (LY2606368) etc. except PF-477736. Is there any reason that the authors select PF-477736 for CHK1 inhibitor in this study? Isn’t there any possibility that PF-477736 is not good for inducing cell death in NB such as NB-39-null and SK-N-BE cells? In that respect, the authors should give additional data compared with another CHK1 inhibitor to check those phenomena would be happened.
2. In several papers, 0.5nM to 540nM concentrations of PF-477736 induced cell death or cytotoxicity through many other mechanisms, even though the cell lines are different. In that respect, 1μM concentration of PF-477736 is too high than other reports. Isn’t it possible that low concentration of PF-477736 is not enough to induce cell death in for NB cell itself? Even though some cell lines such as CHP134 and SMS-SAN were sensitive to 1 μM concentration of PF-477736 rather than
NB-39-nu and SK-N-BE cells, still it seems like 1 μM concentration is too high concentration because some of cell lines show increased viability in 1 μM of it. If it is right, 5 or 10 μM PF-47736 can cause indirect response rather than direct response, because their reactivity is not specific to cell lines.The authors should explain this.
[1] Blasina A, et al. Mol Cancer Ther, 2008, 7(8), 2394-2404.
[2] Chakraborty J, et al. J Biol Chem, 2010, 285(43), 33104-33112.
[3] Zhang C, et al. Clin Cancer Res, 2009, 15(14), 4630-4640.
[4] Carrassa L, et al. Cell Cycle, 2012, 11(13), 2507-2517.
3. As we know, SK-N-BE cells are one of the MYCN-amplified neuroblastoma cell lines. However, as shown in figure 1c, the expression of MYCN is significantly lower in SK-N-BE cells. This is quite different from previously reported papers, and how can the authors explain this difference? Several papers showed the expression levels of MYCN in SK-N-BE cells are higher than this paper. The references are as followings;
(1) Chayka et al., “Identification and Pharmacological Inactivation of the MYCN Gene Network as a Therapeutic Strategy for Neuroblastic Tumor Cells J Biol Chem. 2015 Jan 23;290(4):2198-212.
(2) Human neuroblastoma cells with MYCN amplification are selectively resistant to oxidative stress by transcriptionally up‐regulating glutamate cysteine ligaseJ Neurochem. 2010 May;113(4):819-25
(3) ID2 expression in neuroblastoma does not correlate to MYCN levels and lacks prognostic value Oncogene. 2003 Jan 23;22(3):456-60.
4. Overall, it seems like that there is not enough description for experiments. There should be mentioned how much cells were seeded in each experiment, how the experiments were carried out, and how much different in western blot analysis through statistical graphs. How they set the IC50 in each cell lines etc.
Minor comments but should be addressed.
1. Line 91 : “We found that MYCN-amplified human NB cell lines classified as having low sensitivity to CHK1i activated, the ATM-p53-p21 pathway of the DDR upon CHK1i treatment, thus enabling their survival.” I think that, in order to explain this, the authors should also compare the expression pattern of ATM-P53-P21 in cells with high sensitivity to CHK1i. It may be difficult to understand their assortment by checking with low sensitivity cells only.
2. Line 203; the authors wrote as “whereas CHK1 expression was similar in all four lines (Fig. 1C)”in the text, but I don’t agree with that. In figure 1C, the relative expression does not seem similar because the actin expression level is also different between 4 groups. The authors should add the densitometric analysis of western blot. And SKN-BE as SK-N-BE.
3. Line 208; in figure 1a, the x axis should denote Days. Even though the authors did three repeated experiments, there are no error bars. It should be checked.
4. Line 208; in figure 1A, how did the authors set the O. It should be mentioned in the text. There is no statistical p value.
5. Line 230; on the bar graph, there is no indication for the statistical significance which was mentioned in line 234 as *p<0.05. The author should insert asterisk (*) on the graph.
6. In figure 2A, qRT-PCR analysis showed that P53 expression was lower in the CHK1i-treated group than in Control. However, that result shows a different pattern from western blot result in Figure 2B. How can this be explained?
7. Since the expression pattern of p53 differs from that of Figure 2A and 2B, it is necessary to confirm by additional western blot analysis after treating CHK1i in p53 siRNA treated group in order to prove the clear dependence of P53 as the authors claim.
8. Line 243; How much did CHK1i 5 μM and 10 μM affect viability of NB-39-null cells? The authors performed cell cycle analysis only, but it would be good to compare cell viability with addition of annexin V / PI analysis or MTT and cell morphological analysis with dose-dependent treatment of 1 μM, 5 μM and 10 μM CHK1i treatment.
9. Line 241-254; the authors used CHK1i and PF-477736 together, even though they are the same drug. It seems necessary to unify the terms in the text. Because many CHK1 inhibitors are already used for research, the authors have to use PF-477736 in figures instead of CHK1i.
10. Line 248; If CHK1i activates ATM-p53 signaling by inducing DSB only in insensitive neuroblastoma cell lines with CHK1i, it is necessary to check whether CHK1i induces DSB in neuroblastoma cell lines that are sensitive to CHK1i. It does not seem to give any answer as to whether this is just due to cell diversity. In other words, it is necessary to confirm that changes in γH2AX levels are not found in the other two cells.
11. Line 245~247; “These results indicate that PF-477736 induced DSBs in NB-39-nu cells and triggered DNA repair through the ATM-p53-p21 pathway for cell survival.” How can the authors say treatment of PF-477736 triggered DNA repair? Is there any data for that conclusion? I think there is only increased expression of γH2AX which is a marker of DSBs but not any data about DNA repair. The authors should explain this. Or the authors have to do additional experiments for checking the aneuploidy in this case.
12. Line 249~251;”although exposure of NB-39-nu cells to …. And the DDR was unable to prevent cell death”. This sentence is vague to understand. It should be improved by English editing. And is there any data for DNA damage response?
13. In figure 3B, has the experiment in figure 3B been done only once? It doesn’t have any statistical comparison between each cell cycle as well as any p values.
14. In figure 3B, the authors mentioned that “5 and 10 μM of CHK1i resulted in massive cell death, as detected by the increasing size of the sub-G1 population.” Does it mean that CHK1i (5 and 10 μM) treatment induces sub-G1 arrest and it causes massive cell death? If so, how the author can explain that G2 / M arrest also increased by that treatment in figure 3B?
15. Line 265; “With either the CHK1i at 0.175 μM (below its IC50)”. There is no mention how this concentration was selected. Actually, in cell viability test of figure A, NB-39-Nu cells were not induced cell death at 1 μM concentration. There is no mention at all about the time in which the IC50 is determined and which cells were used to.
16. Line 267; as a method for cell viability test in figure 1A, why the authors didn’t test the alamarBlue assay to check the cell viability? And how the authors calculate relative viable cell numbers? Did you count all cells?
17. In figure 4C, how the cell morphology was changed? The authors have to denote which cells have DNA damage or not. The pictures were too small and it is not able to compare. And how 0.175 μM CHK1i treatment can cause cell death in NB-39-nu cell? In figure 1A, 1 μM CHK1i treatment did not induce any cell death even in 3 days. The authors should explain this difference.
18. Line 285; the authors should add the difference in cell morphology after Gimsa staining in figure 4C and the intensity of western blot in Figure 4B should be indicated.
19. Figure 4B should have densitometric analysis of western blot.
20. Line 290; the authors mentioned as “Giemsa staining and microscopy. The graphs were plotted using the values from four independent experiments”, but where are the graphs? I guess the graphs mean the figure 4A. The authors should revise the figure legend to understand what their results mean exactly.
21. Line 288 & figure 4A; I wonder why the experiment was carried out for 7 days. Of course, the colony formation assay seems to have been performed, if so, the authors have to fill out more detail in method section. In figure 4A, I think that the authors should have treated CHK1i 1μM and ATMi 10 μM for 24 hours or 36 hours as they did before. If the authors have treated the cells for 7 days, does it mean they treated with CHK1i 0.175 μM for 7 days? There is no mention in the method section (line 112-113) about the experimental method of how many cells are seeded first, whether the drug is treated once or for 7 days, and whether the medium is changed every day or not. Also, in line 112, the days are marked as 6days, but in line 288, it is marked as 7days, so the authors are required to check it.
22. Line 298~303; “CHK1 and DNA-PK not only significantly decreased cell viability compared with either single agent alone but also had a greater growth-inhibitory effect than the combination of CHK1i and ATMi (P<0.05, Fig. 5A). …the single agents alone (P<0.05, Fig. 5B).” There is no A and B in figure 5. The authors should check it again. The authors also re-write figure legends consistent with the main text without left/right/upper/lower…
23. In figure 5, it looks like require the densitometric histogram to show their statistical differences.
24. Line 388; In Figure 6, the authors plotted that p21 blocks cdk2 / Cyclin E, but there is no experiment presented in this paper. How can authors assert that without any result of their own? This should be explained and, if necessary, should be revised.
Author Response
We are grateful to reviewer 1 for the critical comments and useful suggestions that have helped us to improve our paper considerably. As indicated in the responses that follow, we have taken all these comments and suggestions into account in the revised version of our paper.
Response to Reviewer 1 Major comments;
Point 1.:There are many other CHK1 inhibitors including Ro-3306, AZD7762, Rabusertib, CHIR-124, MK-8776, GDC-0575 (ARRY-575, RG7741), CCT245737, Prexasertib HCl (LY2606368) etc. except PF-477736. Is there any reason that the authors select PF-477736 for CHK1 inhibitor in this study? Isn’t there any possibility that PF-477736 is not good for inducing cell death in NB such as NB-39-null and SK-N-BE cells? In that respect, the authors should give additional data compared with another CHK1 inhibitor to check those phenomena would be happened.
Response 1.: The reason why we selected PF-477736 as a representative of CHK1 inhibitor is that PF-477736 had beeninvestigated in clinical trials as selective CHK1 inhibitor when we started this study. We agree that we should test other CHK1 inhibitors to see whether combination treatment with CHK1 inhibitor and ATM inhibitor/DNA-PK inhibitor would inhibit the cell growth of CHK1 inhibitor-insensitive neuroblastoma cells. In this respect, we are planning to use the second-generation CHK1 inhibitor prexasertib to investigate further in vitro and in vivo studies as next step.
Point 2.:In several papers, 0.5nM to 540nM concentrations of PF-477736 induced cell death or cytotoxicity through many other mechanisms, even though the cell lines are different. In that respect, 1μM concentration of PF-477736 is too high than other reports. Isn’t it possible that low concentration of PF-477736 is not enough to induce cell death in for NB cell itself? Even though some cell lines such as CHP134 and SMS-SAN were sensitive to 1 μM concentration of PF-477736 rather thanNB-39-nu and SK-N-BE cells, still it seems like 1 μM concentration is too high concentration because some of cell lines show increased viability in 1 μM of it. If it is right, 5 or 10 μM PF-47736 can cause indirect response rather than direct response, because their reactivity is not specific to cell lines. The authors should explain this.
[1] Blasina A, et al. Mol Cancer Ther, 2008, 7(8), 2394-2404.
[2] Chakraborty J, et al. J Biol Chem, 2010, 285(43), 33104-33112.
[3] Zhang C, et al. Clin Cancer Res, 2009, 15(14), 4630-4640.
[4] Carrassa L, et al. Cell Cycle, 2012, 11(13), 2507-2517.
Response 2.: As you kindly showed the above-mentioned reference [1], single agent activity of CHK1 inhibitor (IC50) seems to be broad range in the respective cell lines (e.g. HT29: 1.8μM, Colo205: 1.3μM, PC-3: 1.6μM, MDA-MB-231: 1.4μM and K562: 0.42μM shown in Figure 4). However, we have speculated that cytotoxic effect of CHK1 inhibitor may at least partly be resulted from CHK1 inhibitor-induced DSBs irrespective of combination with a DNA-damaging chemotherapeutic agent or single agent. Thus, in this study, we proposed that reduced concentration of CHK1 inhibitor by combination with ATM inhibitor or DNA-PK inhibitor may reduce the toxic side effects of CHK1i on normal cells. We agree that differential reactivity to CHK1 inhibitor, which is specific to each cell lines have been shown as differential IC50in supplementary Figure S2 as mentioned in revised text at line 204.
Point 3.:As we know, SK-N-BE cells are one of the MYCN-amplified neuroblastoma cell lines. However, as shown in figure 1c, the expression of MYCN is significantly lower in SK-N-BE cells. This is quite different from previously reported papers, and how can the authors explain this difference? Several papers showed the expression levels of MYCN in SK-N-BE cells are higher than this paper. The references are as followings;
(1) Chayka et al., “Identification and Pharmacological Inactivation of the MYCN Gene Network as a Therapeutic Strategy for Neuroblastic Tumor Cells J Biol Chem. 2015 Jan 23;290(4):2198-212.
(2) Human neuroblastoma cells with MYCN amplification are selectively resistant to oxidative stress by transcriptionally up‐regulating glutamate cysteine ligaseJ Neurochem. 2010 May;113(4):819-25
(3) ID2 expression in neuroblastoma does not correlate to MYCN levels and lacks prognostic value Oncogene. 2003 Jan 23;22(3):456-60.
Response 3.: we genuinely considered the reason why there exists a discrepancy of MYCN protein levels between our study and the above-mentioned reference (1), (2) and (3). We realized that MYCN protein levels were compared in four MYCN-amplified cell lines, thus SK-N-BE cells seems to have relatively low MYCN protein among these MYCN-amplified cell lines, especially in our Western-blot detection condition of low exposure. In contrast, these references drew a comparison between MYCN single-copy and MYCN-amplified cell lines, therefore we speculated that increasing exposure of Western-blot detection in the references may causally related with this discrepancy. In addition, we believe that the quality of these cell lines including SK-N-BE cell are enough to use in an experiments, because these cell lines that we used in this study has been reported in several articles from our laboratory (revised text at line 198-199; Islam et al. 2000; Suenaga et al. 2009; Niizuma et al. 2006; Li et al. 2018).
Point 4.: Overall, it seems like that there is not enough description for experiments. There should be mentioned how much cells were seeded in each experiment, how the experiments were carried out, and how much different in western blot analysis through statistical graphs. How they set the IC50 in each cell lines etc.
Response 4.: We seriously took the suggestions. We have described cell numbers that we seeded and also the other details in materials and methods section of revised text at Line 108-117. We have appended normalized intensity of several indistinguishable bands in our western blot in revised Figure 1A and Figure 4B. The IC50data of each cell lines has been added in supplementary Figure S2.
Response to Reviewer 1 Minor comments;
Point 1.:Line 91: “We found that MYCN-amplified human NB cell lines classified as having low sensitivity to CHK1i activated, the ATM-p53-p21 pathway of the DDR upon CHK1i treatment, thus enabling their survival.” I think that, in order to explain this, the authors should also compare the expression pattern of ATM-P53-P21 in cells with high sensitivity to CHK1i. It may be difficult to understand their assortment by checking with low sensitivity cells only.
Response 1.: We agree that we should have showed the differential expression pattern of ATM-P53-P21 pathway between low-sensitive cell lines and high-sensitive cell lines to CHK1 inhibitor, NB-39-nu and SMS-SAN, respectively. We appended the full version of Figure 2B as supplementary Figure S3 and added a sentence in revised text at Line 244-247.
Point 2.:Line 203; the authors wrote as “whereas CHK1 expression was similar in all four lines (Fig. 1C)”in the text, but I don’t agree with that. In figure 1C, the relative expression does not seem similar because the actin expression level is also different between 4 groups. The authors should add the densitometric analysis of western blot. And SKN-BE as SK-N-BE.
Response 2.: We agree that we should performed densitometric analysis of CHK1 expression. As you pointed out, Chk1 expression was relatively lower in NB-39-nu cell line than the others. We have rewritten in the revised text at line 216 and have shown the results in Figure 1C with an explanation in the legend of Figure 1 at line 230. We corrected the name as SK-N-BE in revised text at line 202 and 215.
Point 3.:Line 208; in figure 1a, the x axis should denote Days. Even though the authors did three repeated experiments, there are no error bars. It should be checked.
Response 3.: We agree that x-axisshould be denoted as Days and added error bar in revised Figure 1A.
Point 4.:Line 208; in figure 1A, how did the authors set the O. It should be mentioned in the text. There is no statistical p value.
Response 4.: We agree that we should have described the details how we determined the cell viability in materials and methods section of revised text at Line 108-111. We have shown the statistical significance as asterisks in revised Figure 1A with an explanation of p value to the legend at Line 227.
Point 5.:Line 230; on the bar graph, there is no indication for the statistical significance which was mentioned in line 234 as *p<0.05. The author should insert asterisk (*) on the graph.
Response 5.: We agree that we should indicated asterisks in revised Figure 1A.
Point 6.:In figure 2A, qRT-PCR analysis showed that P53 expression was lower in the CHK1i-treated group than in Control. However, that result shows a different pattern from western blot result in Figure 2B. How can this be explained?
Response 6.: We presume that the results in Figure 2A showed CHK1 inhibitor significantly down-regulated p53 mRNA expression but stabilized p53 in protein level. We agree that we should have described the possible explanation of the data in the revised text at Line239-240 and 248-249.
Point 7.:Since the expression pattern of p53 differs from that of Figure 2A and 2B, it is necessary to confirm by additional western blot analysis after treating CHK1i in p53 siRNA treated group in order to prove the clear dependence of p53 as the authors claim.
Response 7.: We speculate that CHK1 inhibitor significantly down-regulated p53 mRNA expression but upregulated p53 in protein level. Instead of p53 siRNA experiments,the dependency of p53 induced by CHK1 inhibitor was shown in supplementary Figure S3 as the differential response between low-sensitive cell lines and high-sensitive cell lines (NB-39-nu and SMS-SAN, respectively) to CHK1 inhibitor and described the explanation of the data in the revised text at Line 244-249.
Point 8.:Line 243; How much did CHK1i 5 μM and 10 μM affect viability of NB-39-null cells? The authors performed cell cycle analysis only, but it would be good to compare cell viability with addition of annexin V / PI analysis or MTT and cell morphological analysis with dose-dependent treatment of 1 μM, 5 μM and 10 μM CHK1i treatment.
Response 8.: It is worth comparing cell cycle distribution with cell viability to validate whether cell growth inhibition by CHK1 inhibitor could be observed in parallel with an increasing sub-G1 population. We have shown the growth inhibitory effects by CHK1i in supplementary Figure S4 and have modified revised text at Line 272-273. Since Sub-G1 population is commonly recognized as apoptotic cell, we have used the cell distribution analysis as a substitute for cell morphological analysis.
Point 9.:Line 241-254; the authors used CHK1i and PF-477736 together, even though they are the same drug. It seems necessary to unify the terms in the text. Because many CHK1 inhibitors are already used for research, the authors have to use PF-477736 in figures instead of CHK1i.
Response 9.: We unify the terms "CHK1i (PF-477736)" in revised text and the legend of Figures instead of "CHK1i" or "PF-477736" at line 251-274, however the term of CHK1i has used in figures because term of PF-477736 occupies a lot of space of Figures.
Point 10.:Line 248; If CHK1i activates ATM-p53 signaling by inducing DSB only in insensitive neuroblastoma cell lines with CHK1i, it is necessary to check whether CHK1i induces DSB in neuroblastoma cell lines that are sensitive to CHK1i. It does not seem to give any answer as to whether this is just due to cell diversity. In other words, it is necessary to confirm that changes in γH2AX levels are not found in the other two cells.
Response 10.: Previous study by Derenzini et al. (Oncotarget. 2015, 9: 6553-69) has been reported that the differential cellular response to CHK1 inhibitor such as apoptosis or survival might be casually related to the levels of γH2AXaccumulation. In this respect, we have appended γH2AX levelswith or without CHK1 inhibitor in low-sensitive cell lines and high-sensitive cell lines (NB-39-nu and SMS-SAN, respectively) to supplementary Figure S4 and described the explanation of the data including the reference above in revised text at Line 275-277.
Point 11.:Line 245~247; “These results indicate that PF-477736 induced DSBs in NB-39-nu cells and triggered DNA repair through the ATM-p53-p21 pathway for cell survival.” How can the authors say treatment of PF-477736 triggered DNA repair? Is there any data for that conclusion? I think there is only increased expression of γH2AX which is a marker of DSBs but not any data about DNA repair. The authors should explain this. Or the authors have to do additional experiments for checking the aneuploidy in this case.
Response 11.: We agree that "treatment of PF-477736 triggered DNA repair" is likely to be overstatement. Therefore, we have rewritten in revised text at Line 277-280 to describe about it as a possible mechanistic insight.
Point 12.:Line 249~251;”although exposure of NB-39-nu cells to …. And the DDR was unable to prevent cell death”. This sentence is vague to understand. It should be improved by English editing. And is there any data for DNA damage response?
Response 12.: We agree that the sentence does not make sense. Thus, we have rewritten the sentence in revised text at line 279-280 succinctly and clearly.
Point 13.:In figure 3B, has the experiment in figure 3B been done only once? It doesn’t have any statistical comparison between each cell cycle as well as any p values.
Response 13.: We analyzed the cell cycle distribution of each condition by one tube per a condition. We mentioned number of events, 1 x 104events of each condition, in text at line 177.
Point 14.:In figure 3B, the authors mentioned that “5 and 10 μM of CHK1i resulted in massive cell death, as detected by the increasing size of the sub-G1 population.” Does it mean that CHK1i (5 and 10 μM) treatment induces sub-G1 arrest and it causes massive cell death? If so, how the author can explain that G2 / M arrest also increased by that treatment in figure 3B?
Response 14.: We recognized that sub-G1 population was measured as small fragmented DNA of apoptotic cell death. We have mentioned it in revised text at line 272 and included a reference (EAPP: gatekeeper at the crossroad of apoptosis and p21-mediated cell-cycle arrest. Andorfer P and Rotheneder H., Oncogene. 2011 23:2679-90). We concluded that increasing population of G2/M might be causally related to mitotic catastrophe as discussed in the text at line 316-319.
Point 15.:Line 265; “With either the CHK1i at 0.175 μM (below its IC50)”. There is no mention how this concentration was selected. Actually, in cell viability test of figure A, NB-39-Nu cells were not induced cell death at 1 μM concentration. There is no mention at all about the time in which the IC50 is determined and which cells were used to.
Response 15.: Since effect of cell growth inhibition is at least in part related to cell density, we should mentioned the differential cell numbers in revised legends of Figure1A and Figure 4A at line108-109 and 111, respectively. We agree that we should have mentioned IC50data in supplementary Figure S2 with a explanation in the legend to Figure S2 at line 688-692.
Point 16.:Line 267; as a method for cell viability test in figure 1A, why the authors didn’t test the alamarBlue assay to check the cell viability? And how the authors calculate relative viable cell numbers? Did you count all cells?
Response 16.: When we performed cell viability assay showing in Figure 1A, we did not obtain an alamarBlue regent, therefore we counted cell by using Trypan blue exclusion test. We agree that we should have described the details in revised materials and methods section at line 108-111.
Point 17.:In figure 4C, how the cell morphology was changed? The authors have to denote which cells have DNA damage or not. The pictures were too small and it is not able to compare. And how 0.175 μM CHK1i treatment can cause cell death in NB-39-nu cell? In figure 1A, 1 μM CHK1i treatment did not induce any cell death even in 3 days. The authors should explain this difference.
Response 17.: We agree that we should have denoted which cells had DNA damage in revised legend of Figure 4C at line 339-340. We agree that we should mentioned the differential conditions of Figure 1A and Figure 4 in the revised materials and methods at line 108-114. In the respect of cell death, we only observed in cell with combination treatment of CHK1 inhibitor and ATM inhibitor described in the text at Line 316-317.
Point 18.:Line 285; the authors should add the difference in cell morphology after Gimsa staining in figure 4C and the intensity of western blot in Figure 4B should be indicated.
Response 18.: We agree that we should have denoted cell morphology in revised text at line 316-317 and legend of Figure 4C at line 339-340. We appended normalized intensity of several indistinguishable bands of our western blot to revised Figure 4B.
Point 19.:Figure 4B should have densitometric analysis of western blot.
Response 19.:We agree that we should have shown normalized intensity of several indistinguishable bands of our western blot in revised Figure 4B.
Point 20.:Line 290; the authors mentioned as “Giemsa staining and microscopy. The graphs were plotted using the values from four independent experiments”, but where are the graphs? I guess the graphs mean the figure 4A. The authors should revise the figure legend to understand what their results mean exactly.
Response 20.: We agree that we should have modified the sentence in revised legend of Figure 4 at line 337.
Point 21.:Line 288 & figure 4A; I wonder why the experiment was carried out for 7 days. Of course, the colony formation assay seems to have been performed, if so, the authors have to fill out more detail in method section. In figure 4A, I think that the authors should have treated CHK1i 1μM and ATMi 10 μM for 24 hours or 36 hours as they did before. If the authors have treated the cells for 7 days, does it mean they treated with CHK1i 0.175 μM for 7 days? There is no mention in the method section (line 112-113) about the experimental method of how many cells are seeded first, whether the drug is treated once or for 7 days, and whether the medium is changed every day or not. Also, in line 112, the days are marked as 6days, but in line 288, it is marked as 7days, so the authors are required to check it.
Response 21.: We did not performed colony formation assay but alamarBlue cell viability assay in Figure 4A. We should have mentioned "alamarBlue cell viability assay" instead of "cell viability assay" in revised legend of Figure 4A at line 335. As you pointed out, we should have described the details how many cell were seeded as well as how often change the medium in revised materials and methods section at Line 108-117. We have mentioned that alamarBlue cell viability assay and protein expression analysis by immunoblotting at 7 days after indicated treatments, or Giemsa staining and microscopy at 6 days after indicated treatments in revised legend of Figure 4C at line 336-337.
Point 22.:Line 298~303; “CHK1 and DNA-PK not only significantly decreased cell viability compared with either single agent alone but also had a greater growth-inhibitory effect than the combination of CHK1i and ATMi (P<0.05, Fig. 5A). …the single agents alone (P<0.05, Fig. 5B).” There is no A and B in figure 5. The authors should check it again. The authors also re-write figure legends consistent with the main text without left/right/upper/lower…
Response 22.: We agree that we should added A and B in revised Figure 5 and written the legend according to the representative Figure 5A and B at Line 357. We have left "upper/ lower panel" for convenience.
Point 23.:In figure 5, it looks like require the densitometric histogram to show their statistical differences.
Response 23.: Giemsa staining experiment shown in Figure 5 was performed only one time to confirm and visualize the cell viability assay, thus we could not performed densitometric analysis with statistical histograms.
Point 24.:Line 388; In Figure 6, the authors plotted that p21 blocks cdk2 / Cyclin E, but there is no experiment presented in this paper. How can authors assert that without any result of their own? This should be explained and, if necessary, should be revised.
Response 24.: We agree that we did not investigate cdk2 / Cyclin E activity in this paper. Therefore, we have erased cdk2 / Cyclin E in revised Figure 6.
Reviewer 2 Report
In this manuscript, the authors reported the investigation of enhanced efficacy and mechanism of co-inhibition of the DNA damage response and CHK1 in the treatment of neuroblastoma cells, including the action of inducing DSBs and activation of ATM-P53 DDR pathway. Also, they showed improved CHK1i efficacy of the combination treatment on CHK1i-insensitive cells, which is meaningful. The paper is written very well, the rationale is clear, and the data are convincing, except,
1) The reference style is not suitable for International Journal of Molecular Sciences.
2) Figure 1, the error bar is not clear. Figure 6, question marks need to be modified.
In short, this manuscript can be accepted for publication on International Journal of Molecular Sciences after minor revision.
Author Response
We are grateful to reviewer 2 for the useful suggestions that have helped us to improve our paper considerably. As indicated in the responses that follow, we have taken all these comments and suggestions into account in the revised version of our paper.
Point 1) The reference style is not suitable for International Journal of Molecular Sciences.
Response1) We agree that we have changed the reference style to suit for IJMS.
Point 2) Figure 1, the error bar is not clear. Figure 6, question marks need to be modified.
Response2) We agree that we have added the error bar in revised Figure 1A and have modified question marks in Figure 6.
Round 2
Reviewer 1 Report
I am grateful for all the answers and corrections made by the authors. I think that this article has improved considerably more than before. I hope the authors do not feel uncomfortable about many comments I suggested. This paper appears to be sufficient for acceptance.